

# Dihydromyricetin inhibits injury caused by ischemic stroke through the lncRNA SNHG17/miR-452-3p/CXCR4 axis

Jiacheng Xie[1], Qiuyue Yang[1], Xueliang Zeng[2], Qi Zeng[3] and Hai Xiao[1,4]

[1] Department of Pathology, The First Affiliated Hospital of Gannan Medical University, Ganzhou, Jiangxi, China
[2] Department of Pharmacology, The First Affiliated Hospital of Gannan Medical University, Ganzhou, Jiangxi, China
[3] Department of Ultrasound, The First Affiliated Hospital of Gannan Medical University, Ganzhou, Jiangxi, China
[4] Key Laboratory of Prevention and Treatment of Cardiovascular and Cerebrovascular Diseases (Ministry of Education), Gannan Medical University, Ganzhou, Jiangxi, China

Corresponding author
Hai Xiao, 13576675898@163.com

## ABSTRACT

Ischemic stroke (IS) is an important cause of death worldwide. Dihydromyricetin (DHM) has been reported to have neuroprotective potential, but its role and mechanism in IS have not been fully elucidated. The MTT (3-(4,5-dimethylthiazol-2-yl)-2,5-diphenyltetrazolium bromide) assay was used to determine the safe dose of DHM in BV2 microglia and its applicability in OGD/R-treated cells. The mechanism of action of DHM was explored by RT-qPCR, ELISA, luciferase reporter gene assay and western blotting. DHM dose-dependently enhanced BV2 cell viability post-OGD/R and attenuated inflammation and oxidative stress. The protective effects of DHM were found to be mediated through the downregulation of SNHG17, which in turn modulated miR-452-3p expression. miR-452-3p was identified as a negative regulator of pro-inflammatory CXCR4, a direct target whose expression was inversely affected by SNHG17. The interaction between SNHG17 and miR-452-3p was further confirmed by RNA pull-down assays. Furthermore, manipulation of the SNHG17/miR-452-3p/CXCR4 axis was shown to modulate the NF-κB signaling pathway as evidenced by changes in phosphorylation levels. In conclusion, our findings elucidate a novel DHM-mediated neuroprotective mechanism in microglial cells involving the SNHG17/miR-452-3p/CXCR4 regulatory axis. This axis attenuates OGD/R-induced inflammatory and oxidative stress, suggesting a therapeutic potential for DHM in conditions characterized by such pathological processes.

## INTRODUCTION

Ischemic stroke (IS), caused by arterial occlusion, leads to neuronal hypoxia, glucose deprivation, and subsequent brain injury, making it a major cause of disability and mortality worldwide (*Zhu et al., 2022*; *Feske, 2021*; *Shen et al., 2024*). Microglia, the brain's resident immune cells, play a dual role in IS by clearing debris and promoting repair but

also exacerbating neuronal damage through sustained production of pro-inflammatory cytokines and oxidative stress mediators (*Candelario-Jalil, Dijkhuizen & Magnus, 2022*; *Zhu et al., 2021*). Controlling microglial activation and the resulting inflammation is therefore crucial to mitigating IS-induced cerebral injury.

Dihydromyricetin (DHM), a flavonoid extracted from the East Asian herbal vine *Ampelopsis grossedentata*, possesses potent radical scavenging capabilities, suggesting its potential as an anti-inflammatory and antioxidant agent (*Xu et al., 2023*; *Duan et al., 2021*). This potential is supported by numerous studies, including that of *Sun et al. (2021)* which attributed pharmacological actions to DHM, such as anti-inflammatory effects on NF-κB and neuroinflammation, antioxidant properties, and amelioration of mitochondrial dysfunction. In addition, DHM has been found to prevent cerebral aging (*Qian et al., 2021*) and to attenuate LPS-induced neuroinflammatory damage in microglia (*Wei et al., 2022*), but the mechanisms underlying the effects of DHM have not been fully elucidated. The exploration of novel regulatory mechanisms underlying DHM's effects holds significant molecular and clinical implications for advancing its therapeutic potential in the treatment of ischemic stroke.

Small nucleolar RNA host gene 17 (SNHG17) has emerged as an oncogene with upregulated expression in various tumors, such as colorectal (*Bian et al., 2021*) and ovarian (*Zheng et al., 2020*) cancers, promoting tumorigenesis and protecting cancer cells from programmed cell death. Furthermore, SNHG17, as a lncRNA potentially regulating inflammation and oxidative stress, may serve as a critical pathway through which DHM exerts its effects. Studies have suggested that SNHG17 may act as a sponge for downstream miRNA molecules, thereby modulating glioma cell growth (*Ge & Li, 2020*). Additionally, recent studies have indicated that SNHG17 may influence inflammatory pathways by regulating miRNA activity, implicating its potential role in neuroinflammatory conditions. In the context of IS and neuroinflammation, the role of SNHG17 is an emerging area of interest and a focus of the current study. miR-452-3p, a small RNA, has been characterized as an oncogene in hepatocellular carcinoma (*Tang et al., 2017*), but its mechanisms in microglia remain to be fully elucidated. CXCR4, which has been extensively studied in brain disorders, has shown that its antagonists can improve recovery from brain injury (*Friedman-Levi et al., 2021*). By elucidating these interactions, this study not only provides a deeper understanding of DHM's molecular mechanisms but also bridges a critical knowledge gap regarding its role in regulating non-coding RNAs and their downstream inflammatory pathways. However, the interactions between SNHG17 and miR-452-3p as well as between miR-452-3p and CXCR4 have yet to be demonstrated in research.

In the IS scenario, NF-κB serves as a critical molecular link between ischemia-induced cellular stress and the inflammatory response (*Ran et al., 2021*). Loss of perfusion or oxygenation followed by reperfusion initiates the production of reactive oxygen species (ROS) and other stress signals capable of activating NF-κB (*Li, Sun & Jiang, 2021*). Upon activation, NF-κB translocates to the nucleus and promotes the expression of a variety of genes, including those encoding adhesion molecules, chemokines, cytokines, and enzymes involved in the inflammatory response. These NF-κB-mediated pathways significantly promote the expansion of ischemic injury by promoting inflammation, apoptosis, and

pyroptosis within brain tissue (*Liao et al., 2020*; *Luo et al., 2020*). As a fundamental modulator of immune and inflammatory responses, delineating the exact role of NF-κB in the pathophysiology of IS is imperative for the development of targeted treatment modalities that mitigate injury and enhance recovery while minimizing potential adverse effects.

Essentially, our research focuses on the neuroprotective properties of DHM, along with the regulatory functions of lncRNAs and miRNA within microglia under oxygen-glucose deprivation and reoxygenation (OGD/R) conditions. We hypothesize that DHM may exert its influence by modulating the SNHG17/miR-452-3p/CXCR4 axis, thereby affecting microglial inflammatory and oxidative stress responses, potentially paving the way for novel therapeutic avenues for neuroinflammatory diseases. This study addresses a critical knowledge gap by providing mechanistic insights into DHM's regulatory pathways, further underscoring its potential therapeutic significance.

## METHODS

### Cell culture and treatment

The BV2 and HEK293 cell lines (Procell, Wuhan, China) was maintained in Dulbecco's Modified Eagle's Medium (DMEM; Gibco, Thermo Fisher Scientific, Waltham, MA, USA) supplemented with 10% fetal bovine serum at 37 °C. To replicate ischemic stroke (IS) conditions *in vitro*, an oxygen-glucose deprivation and reoxygenation (OGD/R) model was employed. In order to induce OGD, the cells were subjected to a glucose-free DMEM and thereafter incubated within a hypoxic chamber with an atmosphere consisting of 1% $O_2$, 94% $N_2$, and 5% $CO_2$ for a duration of 4 h. After OGD, the cells were returned to normal culture conditions (DMEM with 10% fetal bovine serum in a humidified incubator with 5% $CO_2$) for reoxygenation for 24 h. Concurrently, DHM (Sigma-Aldrich, St. Louis, MO, USA) was administered to the cells at varying concentrations (1, 5, 10 μM) for 24 h prior to OGD/R treatment, as previous research described (*Xie et al., 2022*; *Zhang et al., 2021*), and its molecular structure is shown in Fig. 1.

### Transfection experiments

BV2 cells were transfected with SNHG17 overexpression vector (ov-SNHG17), miR-452-3p mimic/inhibitor, and three small interfering RNA (siRNA) against CXCR4, SNHG17 (si-SNHG17), and their negative controls (NCs) using Lipofectamine® 3000. Their sequence is shown in Table 1.

### Cell viability assessment

Following treatment with OGD/R and/or DHM at concentrations of 1, 5, 10, 20, and 40 μM, the impact of DHM on cellular viability was assessed using the MTT (3-(4,5-dimethylthiazol-2-yl)-2,5-diphenyltetrazolium bromide) test. In summary, cellular entities were introduced into 96-well plates and subjected to the prescribed treatment protocol. After the completion of the treatment, MTT solution (Beyotime, Shanghai, China) was introduced into each well, and the plates were subjected to a 4-h incubation period. Subsequently, the medium was disposed of, and dimethyl sulfoxide (DMSO) from

**Figure 1 Chemical structure of DHM.**

**Table 1 Sequences of the RNAs used in this study.**

| Gene | Sequence |
| --- | --- |
| si-CXCR4-1 | ACAAATGTACAGTCTTGTATT |
| si-CXCR4-2 | GAATGTGTGGTAAATTGAATT |
| si-CXCR4-3 | GTGGATGGTGGTGTTTCAATT |
| si-SNHG17 | GCAUUGGUUACUAUACAAATT |
| si-NC | AATAATCCTGTATGTTATGAC |
| Mimic NC | GTGTAATACACTCGCTTACGAG |
| MiR-452-3p mimic | TCAGTCTCATCTGCAAAGAGGT |
| Inhibitor NC | TATGCACTTATGAAACTCGCGG |
| MiR-452-3p inhibitor | ACCTCTTTGCAGATGAGACTGA |

Sigma-Aldrich was introduced to facilitate the dissolution of the formazan crystals. The measurement of absorbance was conducted at a wavelength of 450 nm using a microplate reader manufactured by Thermo Fisher Scientific.

## Intracellular ROS level evaluation

To evaluate intracellular ROS levels, BV2 cells were seeded at 96-well plate follow by OGD/R or DHM treatment. BV2 cells were then incubated with 2′,7′-dichlorodihydrofluorescein diacetate (Beyotime, Shanghai, China) probe for 15 min. After washing the excess dye with PBS, the cells were collected, the fluorescence was measured using a microplate reader, as described in previous study (*Farkhondeh et al., 2021*).

## Enzyme-linked immunosorbent assays (ELISA)

To quantify the levels of inflammatory cytokines, tumor necrosis factor-alpha (TNF-α), interleukin (IL)-6, IL-1β, and IL-18, along with oxidative stress-related factors such as malondialdehyde (MDA), glutathione (GSH), and superoxide dismutase (SOD), ELISA were performed. All kits are purchased from Solarbio (Beijing, China). Briefly, samples containing these biomarkers were applied to pre-coated ELISA plates specific to each target. After a series of binding, washing, and visualization steps, which included the addition of a substrate that reacts with the enzyme-linked detection antibodies, the intensity of the developed color, proportional to the concentration of the target molecules

in the samples, was quantitatively measured using a spectrophotometer (*Wang et al., 2021*).

### TUNEL assays

BV2 cells were collected and washed twice with cold phosphate-buffered saline (PBS) before being resuspended in binding buffer at a concentration of approximately $1 \times 10^6$ cells/ml. Next, After washing with PBS, sections were incubated with TUNEL reaction mixture in a humidified chamber at 37 °C for 60 min. Sections were then washed three times with PBS and counterstained with DAPI for 5 min to visualize cell nuclei.

### The phenomenon of nucleocytoplasmic separation and RT-qPCR assays

RNA isolation from BV2 cells was performed using the Cytoplasmic and Nuclear RNA Purification Kit (Norgen Biotek, Ontario, Canada). The extraction of total RNA from BV2 cells was performed using TRIzol reagent (Invitrogen, Waltham, MA, USA). The synthesis of cDNA was performed using 1 μg of total RNA and the PrimeScript RT Reagent Kit (Takara Bio, Shiga, Japan). The PCR reactions were conducted with the SYBR Green PCR Master Mix (Applied Biosystems, Thermo Fisher Scientific, Waltham, MA, USA) on a StepOne Real-Time PCR System (Applied Biosystems, Waltham, MA, USA). The thermal cycling protocol consisted of an initial denaturation step at a temperature of 95 °C for a duration of 30 s, followed by 40 cycles of denaturation at 95 °C for 15 s, annealing at 60 °C for 30 s, and extension at 70 °C for 30 s. The RNA levels of long non-coding RNA SNHG17, CXCR4, miR-452-3p, NLRP3, ASC, Caspase 1, and GSDMD were determined using the $2^{-\Delta\Delta Ct}$ method, with GAPDH or U6 serving as internal reference genes for normalization. The primer sequences for these genes are listed in Table 2.

### RNA pull-down assay

Biotinylated SNHG17 (bio-SNHG17), bio-SNHG17 mut, bio-NC, bio-miR-452-3p, bio-miR-452-3p mut, and bio-miR-NC were introduced into BV2 cells by transfection. As per the guidelines provided by the manufacturer, the cell lysates were subjected to a 30-min incubation period at a temperature of 21 °C, during which they were gently agitated, in the presence of Dynabeads M-280 streptavidin (Invitrogen, Waltham, MA, USA). Quantitative analysis was conducted using RT-qPCR.

### Bioinformatic analysis

Potential downstream targets of SNHG17 and miR-452-3p were predicted using bioinformatics tools such as starBase (https://rnasysu.com/encori/), TargetScan 8.0 (https://www.targetscan.org/vert_72/), TarBase v9.0 (https://dianalab.e-ce.uth.gr/tarbasev9), and miRDB (http://www.mirdb.org/).

### Luciferase reporter assay

The psiCHECK-2 dual luciferase reporter vector (Promega Corp., Madison, WI, USA) was utilized to clone SNHG17 and 3′-UTR segments of CXCR4 that contain binding sites for miR-452-3p. The BV2 cell line was subsequently subjected to cotransfection with the

**Table 2 Primer sequences used for RT-qPCR analysis.**

| Primers | Forward primer 5′-3′ | Reverse primer 5′-3′ |
|---|---|---|
| miR-299a-3p | ACACTCCAGCTGGGUAUGUGGGACGGUAAA | CTCAACTGGTGTCGTGGA |
| miR-452-3p | ACACTCCAGCTGGGUCAGUCUCAUCUGCAA | CTCAACTGGTGTCGTGGA |
| miR-3074-5p | ACACTCCAGCTGGGGUUCCUGCUGAACUGA | CTCAACTGGTGTCGTGGA |
| miR-467a-3p | ACACTCCAGCTGGGCAUAUACAUACACACA | CTCAACTGGTGTCGTGGA |
| miR-574-5p | ACACTCCAGCTGGGUGAGUGUGUGUGUGUGA | CTCAACTGGTGTCGTGGA |
| U6 | CTCGCTTCGGCAGCACA | AACGCTTCACGAATTTGCGT |
| SNHG17 | CATTGCCATCCGCCCAAATC | CCTGGTCAATGGATTCGCCT |
| CXCR4 | GTGCAGCAGGTAGCAGTGAA | CCATGGCAACACTCGCTCTA |
| NLRP3 | TCTGTTCATTGGCTGCGGAT | TAGCCGCAAAGAACTCCTGG |
| ASC | GGACAGTTACCAGGCAGTTCG | GTCACCAAGTAGGGCTGTGT |
| Caspase 1 | GACCGAGTGGTTCCCTCAAG | GACGTGTACGAGTGGGTGTT |
| GSDMD | ATGCCATCGGCCTTTGAGAAA | AGGCTGTCCACCGGAATGA |
| GAPDH | GGTGAAGGTCGGTGTGAACG | CTCGCTCCTGGAAGATGGTG |

reporter vector and miR-452-3p mimics for a duration of 48 h, employing Lipofectamine® 3000 as the transfection reagent. The measurement of luciferase activity was conducted with a dual-luciferase reporter assay kit provided by Promega. The firefly luciferase data were normalized using the ratio of firefly to Renilla luciferase activity.

## Western blotting

The denatured proteins were separated using a 10% SDS-PAGE gel (Solarbio, Beijing, China). Subsequently, they were transferred onto a polyvinylidene fluoride membrane and blocked with 5% bovine serum albumin (Solarbio, Beijing, China). The membrane was then incubated overnight at 4 °C with primary antibodies including CXCR4 (1:1,000; ab181020; Abcam, Cambridge, UK), NF-κB (1:1,000; ab207297; Abcam, Cambridge, UK), p-NF-κB (1:1,000; ab239882; Abcam, Cambridge, UK), and GAPDH (1:5,000; ab181602; Abcam, Cambridge, UK). Subsequently, the sections underwent two rinses with TBST buffer (Solarbio, Beijing, China; containing 0.05% Tween 20) for a duration of 10 min each. Following this, the sections were incubated with goat anti-rabbit IgG H&L (1:12,000; ab205718; Abcam, Cambridge, UK) for a period of 1 h at a temperature of 21 °C.

## Statistical analysis

The data are reported in the form of the mean value plus or minus the standard deviation, all experiment was repeated three times. The dissimilarities across groups were examined using the application of one-way analysis of variance (ANOVA) followed by Bonferroni *post hoc* analysis. A *p*-value that is less than 0.05 was deemed to be statistically significant.

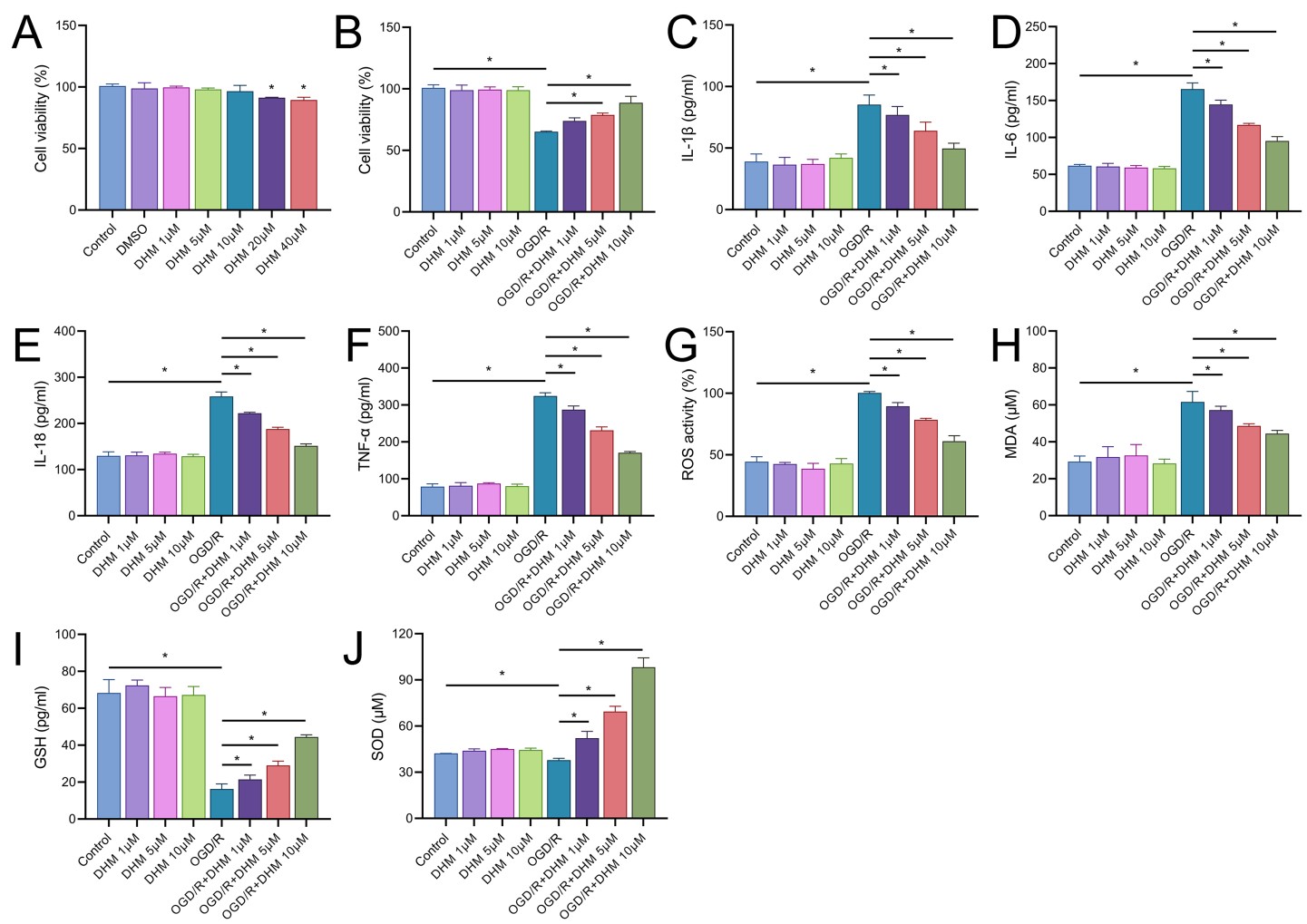

**Figure 2 DHM modulates cell viability and stress responses in BV2 cells.** (A) MTT assay showing the cytotoxic effects of varying concentrations of DHM on BV2 cells. (B) MTT assay results depicting cell viability after pretreatment with non-toxic DHM concentrations prior to OGD/R. (C–J) ELISA analyses of inflammatory cytokines (C–E) and oxidative stress markers (F–J), including ROS and MDA levels along with SOD activity, in response to OGD/R and subsequent DHM treatment. *$P < 0.05$.

# RESULTS

## SNHG17 is an anti-inflammatory and antioxidant pathway exerted by DHM

We commenced our study by assessing the cytotoxic effects of DHM in BV2 cells. The MTT assay delineated that cell viability was compromised at DHM concentrations of 20 μM or greater. Consequently, we utilized DHM at concentrations of 1, 5, and 10 μM for subsequent experimentation (Fig. 2A). Pretreatment of BV2 cells with these concentrations prior to OGD/R revealed a significant promotion of cell viability under these non-toxic DHM concentrations (Fig. 2B). In addition, post-OGD/R BV2 cells exhibited increased levels of inflammatory cytokines (IL-1β, IL-6, IL-18, TNF-α) and oxidative stress markers (ROS, MDA), alongside reduced SOD activity. DHM dose-dependently mitigated these changes (Figs. 2C–2J).

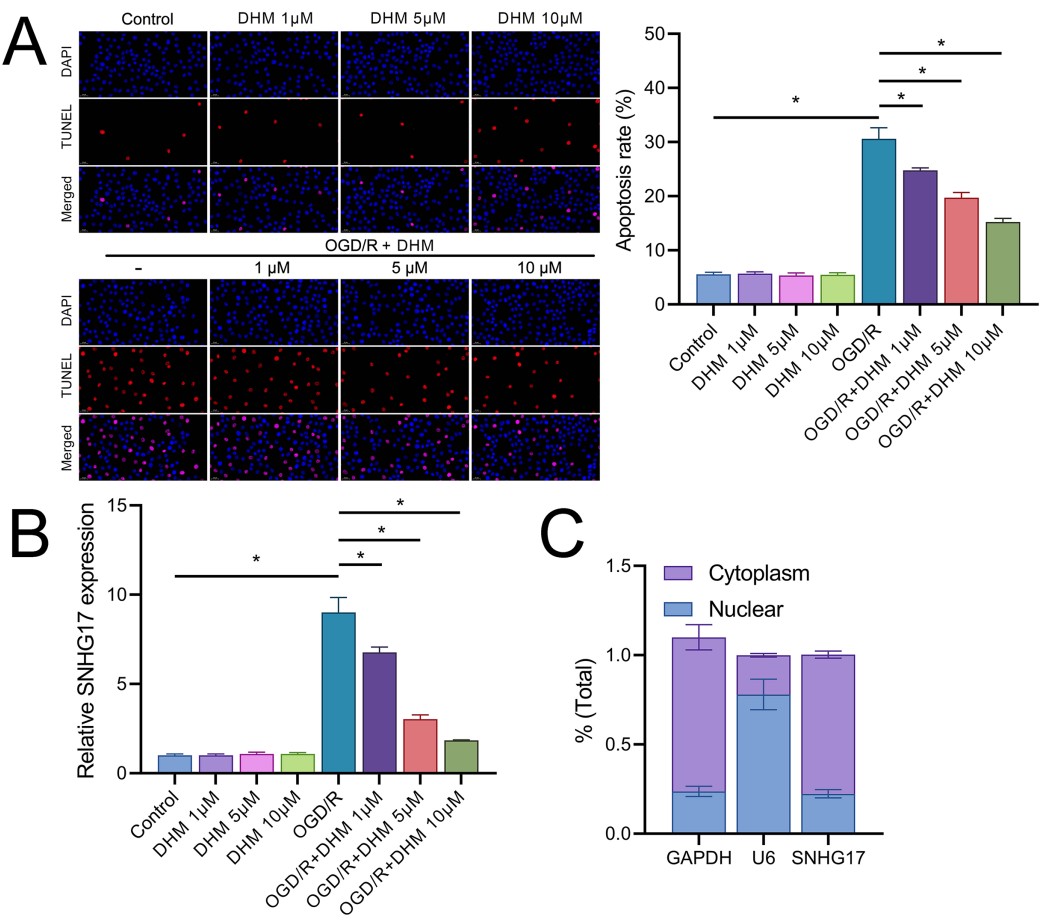

**Figure 3 DHM modulates cell apoptosis in BV2 cells.** (A) TUNEL analysis of apoptosis in response to OGD/R and subsequent DHM treatment. (B) RT-qPCR analysis of SNHG17 expression post-OGD/R, illustrating dose-dependent suppression by DHM. (C) Nuclear-cytoplasmic fractionation assay determining the cellular localization of SNHG17. *$P < 0.05$.

In addition, DHM reduced OGD/R-induced apoptosis in a dose-dependent manner (Fig. 3A). In this context, we observed that the expression of SNHG17 respond to inflammatory, oxidative stress, and apoptosis with a significant upregulation post-OGD/R which was dose-dependently suppressed by DHM (Fig. 3B). Nuclear-cytoplasmic fractionation assays showed that SNHG17, congruent with the positive control GAPDH and in contrast to U6, was predominantly expressed in the cytoplasm and less so in the nucleus (Fig. 3C), suggesting that this lncRNA might modulate downstream targets and signaling *via* the ceRNA mechanism.

## SNHG17 directly targets and binds miR-452-3p

Vectors for the overexpression and silencing of SNHG17 were constructed and transfected into BV2 cells. Relative to their respective controls, the expression of SNHG17 was augmented in the ov-SNHG17 group and diminished in the si-SNHG17 group, with the efficacy of these vectors confirmed by RT-qPCR (Fig. 4A). MTT assays indicated that cell viability reduction by OGD/R in BV2 cells was further suppressed by ov-SNHG17 but was

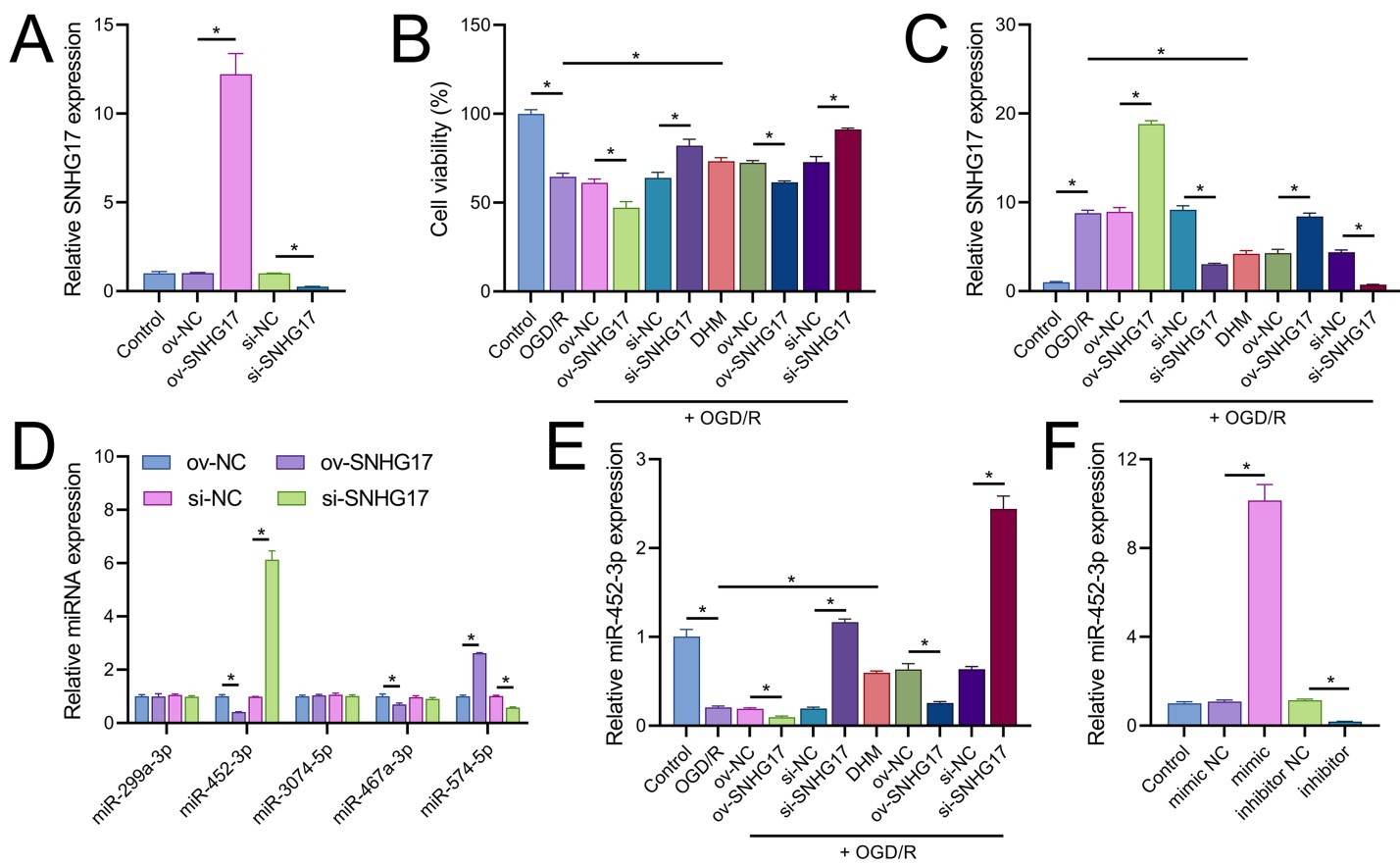

**Figure 4 SNHG17 interaction with miR-452-3p in BV2 cells.** (A) RT-qPCR analysis of SNHG17 expression after transfection with vectors for overexpression (ov-SNHG17) and silencing (si-SNHG17). (B) MTT assay evaluating the impact of SNHG17 modulation on BV2 cell viability following OGD/R. (C) RT-qPCR confirmation of SNHG17 expression changes post-DHM treatment and vector transfection. (D) RT-qPCR analysis identifying miRNAs with potential binding sites to SNHG17 and (E) the regulation of RT-qPCR analysis of SNHG17 regulation of miR-299a-3p, miR-452-3p, miR-3074-5p, miR-467a-3p, and miR-574-5p by SNHG17. (F) RT-qPCR validation of miR-452-3p mimic and inhibitor efficacy. *$P < 0.05$.

enhanced by si-SNHG17; the partial mitigation of OGD/R-induced viability decrease by DHM was lessened by ov-SNHG17 but was increased by si-SNHG17 (Fig. 4B). The elevation of SNHG17 expression by OGD/R was partially offset by DHM, and the level of SNHG17 was increased by ov-SNHG17 and decreased by si-SNHG17 (Fig. 4C), corroborating that DHM's action on OGD/R was mediated through SNHG17. Through the starBase database, we identified 21 miRNAs with potential binding sites to SNHG17. After excluding those miRNAs with established mechanistic studies in brain diseases *via* PubMed, five remained: miR-299a-3p, miR-452-3p, miR-3074-5p, miR-467a-3p, and miR-574-5p. MiR-452-3p expression was negatively regulated by SNHG17 (Fig. 4D). The remaining four miRNAs (miR-299a-3p, miR-3074-5p, miR-467a-3p, and miR-574-5p) were excluded from further analysis because they either showed no significant changes in the presence of ov-SNHG17 or si-SNHG17 (miR-299a-3p, miR-3074-5p, miR-467a-3p) or exhibited a positive correlation with SNHG17 expression (miR-574-5p), which is inconsistent with the ceRNA mechanism of SNHG17. Further analysis demonstrated its

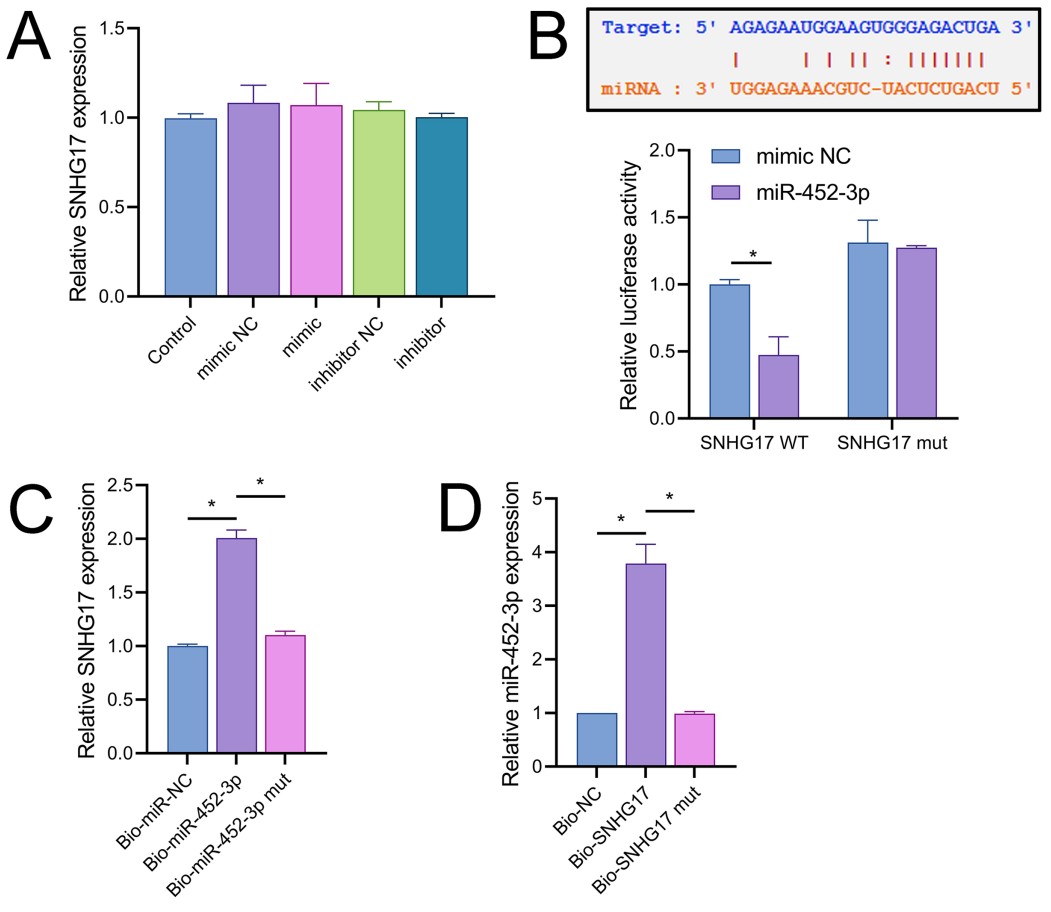

**Figure 5 SNHG17 negatively regulates miR-452-3p in BV2 cells.** (A) RT-qPCR showing expression changes of SNHG17 after miR-452-3p mimic/inhibitor transfection. (B) Dual-luciferase reporter assay confirming the direct interaction between SNHG17 and miR-452-3p. (C, D) RNA pull-down assays demonstrating the binding of SNHG17 and miR-452-3p. *$P < 0.05$.

reduced expression post-OGD/R, which was recoverable with DHM pretreatment (Fig. 4E). RT-qPCR confirmed the effectiveness of miR-452-3p mimics and inhibitors, showing increased expression in the mimic group and decreased expression in the inhibitor group (Fig. 4F).

Notably, miR-452-3p expression changes did not significantly alter SNHG17 levels (Fig. 5A). Dual-luciferase reporter assays showed that luminescence activity was significantly decreased in cells cotransfected with WT-SNHG17 (47.33 ± 13.58% *vs* 100.00 ± 3.61%), and the mimic compared to the mimic NC, with no notable difference in luminescence activity between mimic NC and mimic groups (127.33 ± 1.53% *vs* 131.33 ± 16.65%) when cotransfected with mut-SNHG17 (Fig. 5B). Moreover, RNA pull-down assays exhibited marked enrichment of SNHG17 or miR-452-3p in the bio-miR-452-3p or bio-SNHG17 as opposed to the negative control and mut groups (Figs. 5C, 5D). These findings substantiate that SNHG17 directly targets and binds to miR-452-3p.

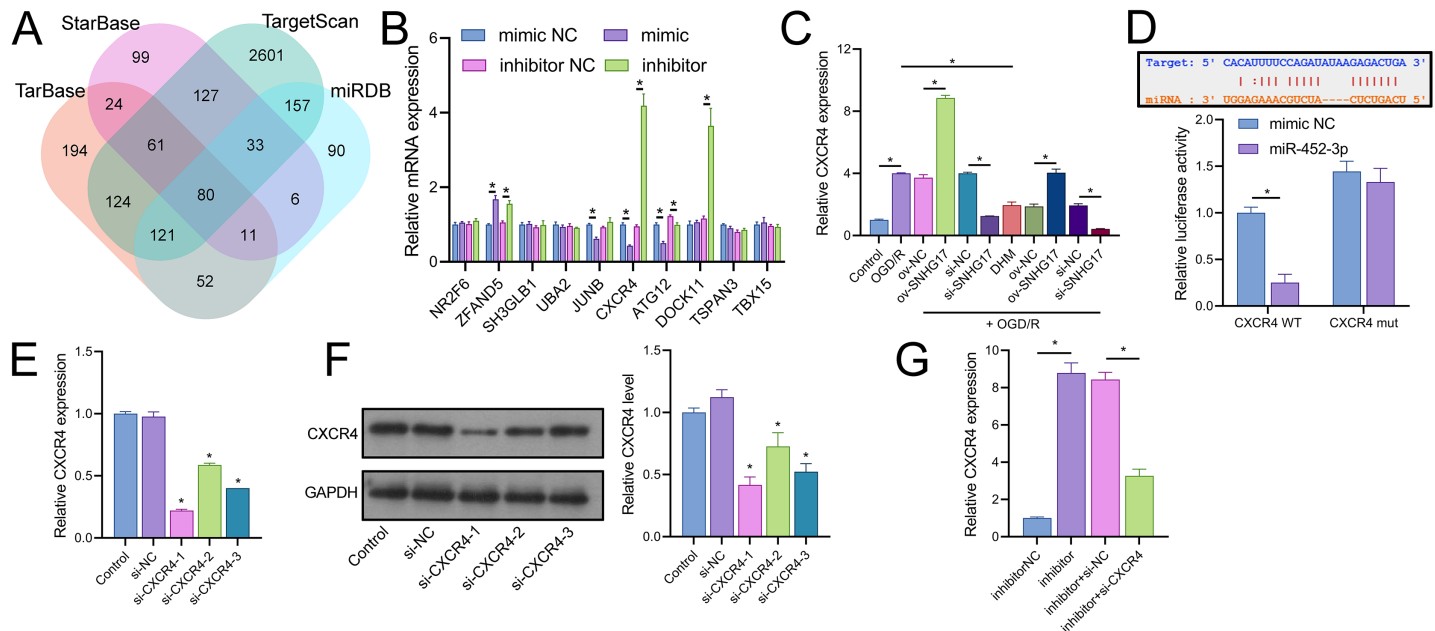

**Figure 6 Identification of CXCR4 as a direct target of miR-452-3p.** (A) Venn diagram showing Targetscan, miRDB, and starBase databases jointly analyzing downstream targets of miR-452-3p. (B) RT-qPCR analysis of the top 10 gene candidates to identify those negatively regulated by miR-452-3p. (C) RT-qPCR demonstrating the modulation of CXCR4 expression by OGD/R and DHM treatment. (D) Dual-luciferase reporter assay validating CXCR4 as a direct target of miR-452-3p. (E, F) RT-qPCR and western blot analyses determining the efficiency of siRNAs targeting CXCR4. (G) RT-qPCR showing the effects of miR-452-3p inhibitor on CXCR4 expression, and its alteration by si-CXCR4. *$P < 0.05$.

## CXCR4 is a direct target of miR-452-3p

Joint analysis through Targetscan, miRDB, TarBase, and starBase databases identified 80 potential targets of miR-452-3p (Fig. 6A). RT-qPCR of the top 10 Targetscan-predicted genes identified CXCR4 as the sole gene negatively regulated by miR-452-3p (Fig. 6B). CXCR4 was positively regulated by SNHG17, and its expression, elevated in OGD/R-treated cells, was reduced by DHM treatment (Fig. 6C). Dual-luciferase reporter assays indicated that luminescence activity was reduced in cells cotransfected with WT-CXCR4 and the mimic compared to the mimic NC (25.00 ± 14.73% *vs* 100.00 ± 11.02%); no difference was observed in luminescence activity between mimic NC and mimic groups when cotransfected with mut-CXCR4 (133.00 ± 14.73% *vs* 144.33.00 ± 11.02%) (Fig. 6D). Subsequently, we validated three siRNAs targeting CXCR4, with RT-qPCR and western blot analyses indicating the highest efficiency of inhibition with si-CXCR4-1 (RT-qPCR: si-CXCR4-1, si-CXCR4-2, si-CXCR4-2 *vs* si-NC, 22.53 ± 1.02%, 60.70 ± 1.02%, 40.96 ± 0.00% *vs* 97.67 ± 3.79%; western blot: si-CXCR4-1, si-CXCR4-2, si-CXCR4-2 *vs* si-NC, 37.09 ± 5.72%, 64.69 ± 9.77%, 46.59 ± 5.79% *vs* 112.33 ± 6.03%), which was therefore selected for subsequent mechanistic studies (Figs. 6E, 6F). Additionally, the miR-452-3p inhibitor-induced enhancement of CXCR4 expression was abrogated by si-CXCR4 (Fig. 6G). These results confirm that CXCR4 is a direct target of miR-452-3p.

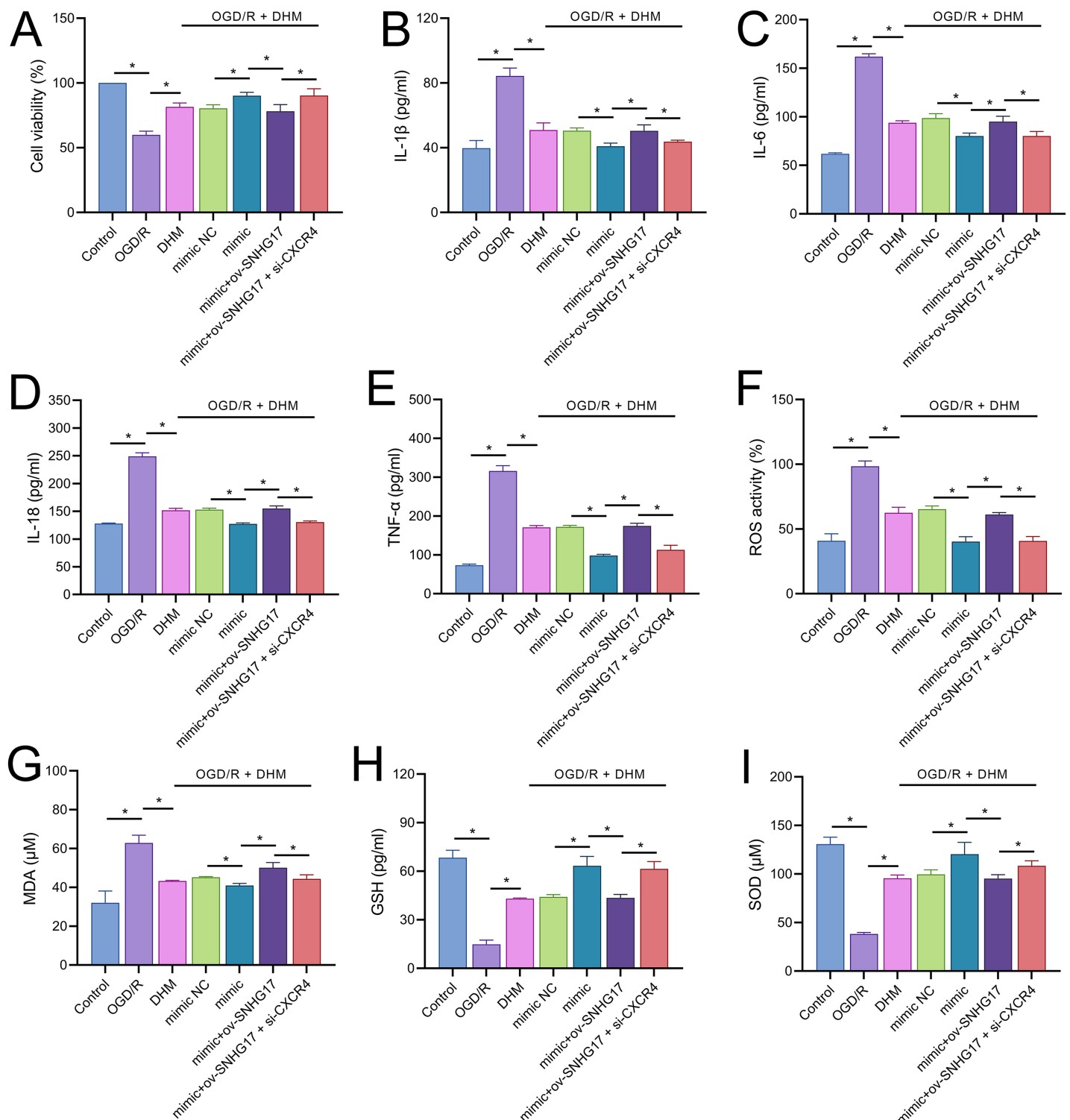

**Figure 7 DHM's protective mechanism in OGD/R-induced damage through the SNHG17/miR-452-3p/CXCR4/NF-κB axis.** (A) MTT assay depicting the effects of the SNHG17/miR-452-3p/CXCR4 axis on cell viability in DHM-treated, OGD/R-challenged BV2 cells. (B–I) ELISA analyses of inflammatory and oxidative stress markers illustrating the modulation by miR-452-3p mimic, ov-SNHG17, and si-CXCR4. *P < 0.05.

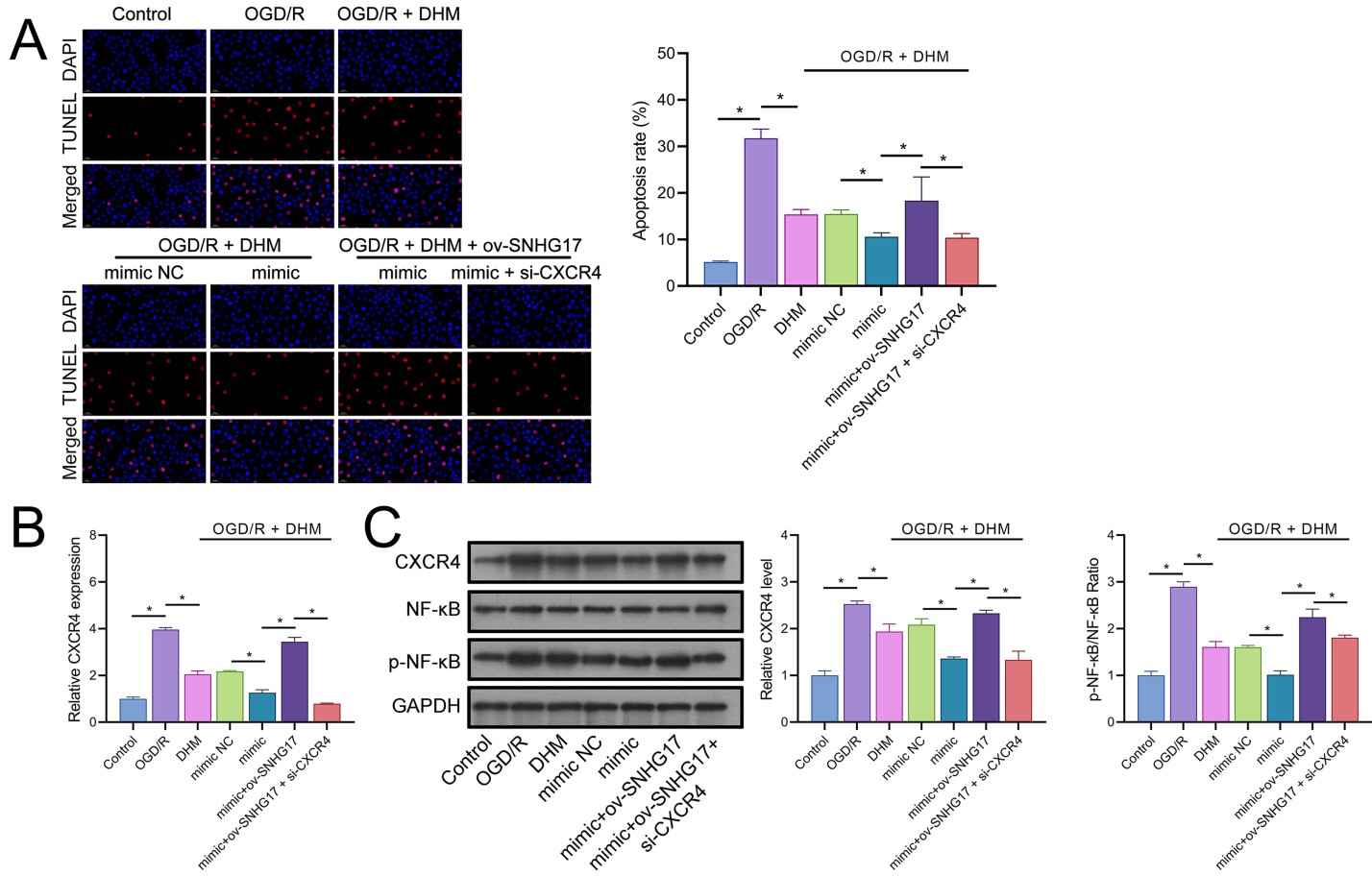

**Figure 8 DHM's protective mechanism in OGD/R-induced damage through the SNHG17/miR-452-3p/CXCR4/NF-κB axis.** (A) TUNEL analysis the effects of the SNHG17/miR-452-3p/CXCR4 axis on apoptosis in DHM-treated, OGD/R-challenged BV2 cells. (B) RT-qPCR analysis of CXCR4 expression post-transfection and DHM treatment. (C) Western blot analysis showing the influence of miR-452-3p, ov-SNHG17, and si-CXCR4 on CXCR4, NF-κB, and p-NF-κB phosphorylation levels. *$P < 0.05$.

## DHM attenuates OGD/R-induced damage *via* SNHG17/miR-452-3p/CXCR4/NF-κB

To validate the mechanism of the SNHG17/miR-452-3p/CXCR4 axis in DHM-treated OGD/R-challenged BV2 cells, we cotransfected miR-452-3p mimic, ov-SNHG17, and si-CXCR4 into cells pre-treated with DHM. MTT results indicated that the miR-452-3p promoted, while ov-SNHG17 counteracted the DHM-induced enhancement of cell viability post-OGD/R, which was reversed by si-CXCR4 (Fig. 7A). DHM mitigated OGD/R-induced inflammatory cytokines and oxidative stress, effects enhanced by miR-452-3p, reversed by ov-SNHG17, and restored by si-CXCR4 (Figs. 7B–7I).

The enhanced effect of miR-452-3p on DHM in inhibiting Bv2 cell apoptosis was offset by the overexpression of SNHG17, which was reversed with the intervention of si-CXCR4 (Fig. 8A). The DHM-mediated upregulation of CXCR4 expression was promoted by miR-452-3p, counteracted by ov-SNHG17, but reversed by si-CXCR4 (Fig. 8B). Western blot analysis indicated that miR-452-3p further elevated the promotion of CXCR4 and NF-κB

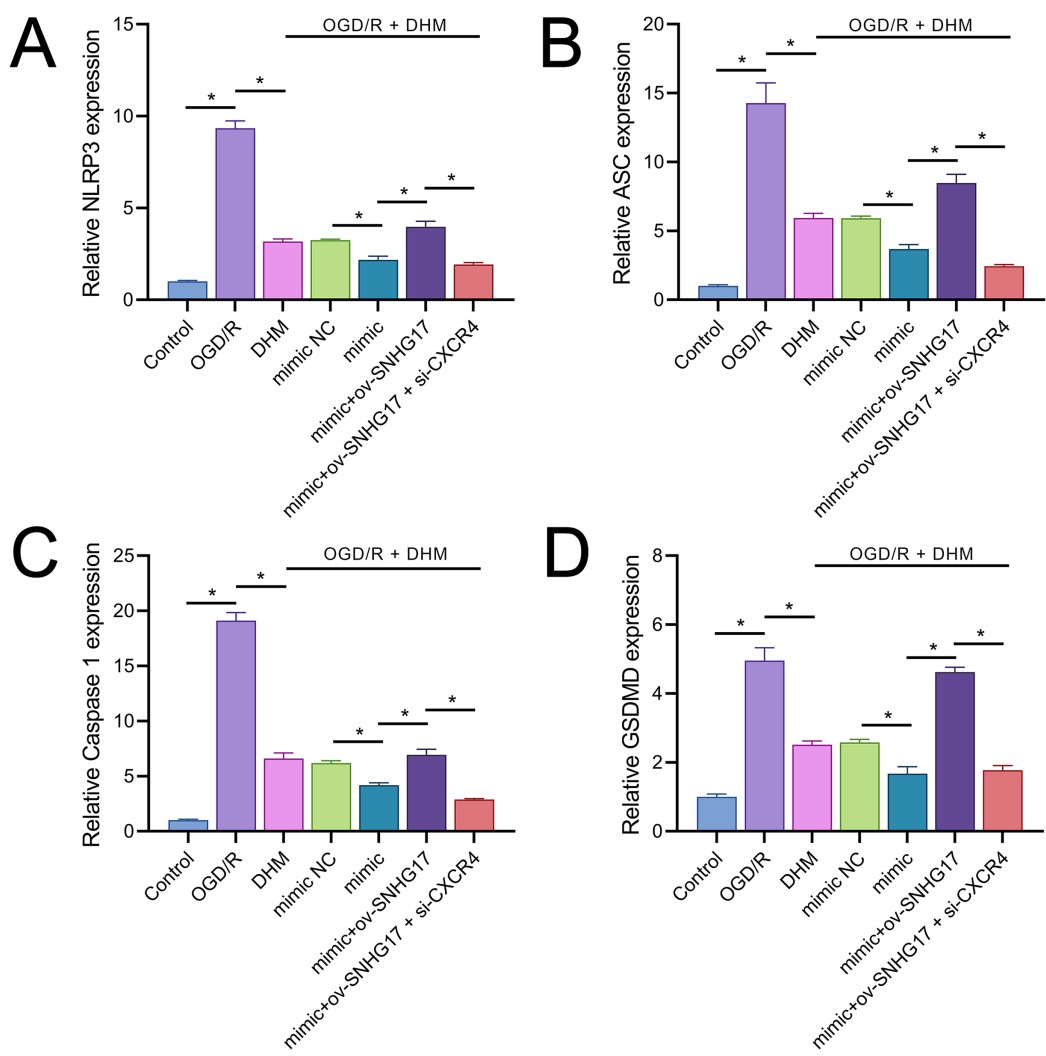

**Figure 9 DHM's modulation of pyroptosis-related markers in OGD/R conditions.** (A–C) RT-qPCR analysis and quantification of inflammasome components including NLRP3, ASC, and Caspase 1 in OGD/R-treated BV2 cells. (D) RT-qPCR demonstrating the effect of DHM on GSDMD expression in cells subjected to OGD/R. *$P < 0.05$.

phosphorylation levels by DHM, which was neutralized by ov-SNHG17 but reversed by si-CXCR4 (Fig. 8C).

## DHM attenuates pyroptosis induced by OGD/R through the SNHG17/ miR-452-3p/CXCR4 axis

Owing to the significant roles of IL-1β and IL-18, we analyzed the expression of inflammasomes. The results demonstrated that OGD/R notably activated inflammasomes, with significant upregulation in the expression of key components NLRP3, ASC, and Caspase 1 (Figs. 9A–9C). Additionally, OGD/R also promoted the expression of the pyroptosis-related gene GSDMD (Fig. 9D). These effects were alleviated by DHM, amplified by miR-452-3p, counteracted by overexpressed SNHG17, and restored by si-CXCR4.

## DISCUSSION

This study concentrated on elucidating the modulatory effects and mechanisms of DHM on microglial cells subjected to OGD/R, with an emphasis on the principles behind the onset and clearance of inflammatory and oxidative stress responses. Given the rapid activation of microglia post-ischemia, these cells fulfill a dichotomous role; on one hand, they facilitate neural regeneration and repair by clearing necrotic neurons and synaptic debris and releasing neurotrophic factors. On the other hand, the chronic activation of microglia results in the excessive secretion of pro-inflammatory cytokines and chemokines, exacerbating neuroinflammation and neuronal injury (*Zhang et al., 2022*; *Zheng et al., 2022*). Hence, comprehending the activation and regulatory mechanisms of microglia is of significant importance for developing therapeutic strategies for IS.

In our study, DHM emerges as a potential neuroprotective agent, capable of significantly reducing the production of inflammatory mediators in BV2 cells treated with OGD/R. This effect appears to be mediated through the enhanced secretion of GSH and SOD, which facilitate the clearance of accumulated levels of ROS and MDA. In comparison to other antioxidant and anti-inflammatory agents, DHM offers the advantage of dual regulatory effects on both oxidative stress and inflammatory pathways, specifically through its modulation of the SNHG17/miR-452-3p/CXCR4 axis, a mechanism not commonly targeted by other compounds. Previous research has substantiated DHM's cytoprotective effects in OGD/R models (*Xie et al., 2022*; *Zhang et al., 2021*). Furthermore, DHM's anti-inflammatory and antioxidant properties have been corroborated in recent studies (*Wang et al., 2023*; *Chen et al., 2023*), aligning with our findings and bolstering the hypothesis that DHM mitigates microglia-mediated neuroinjury by influencing specific inflammation and oxidative stress-related pathways. Against this backdrop, we explored the interplay of SNHG17 with miR-452-3p and CXCR4. Long non-coding RNAs such as SNHG17 play critical roles in regulating cellular processes, including proliferation, apoptosis, and inflammation (*Zhang et al., 2023*; *Han et al., 2020*). Our data suggest that under OGD/R conditions, DHM mitigated the reduction in cellular viability, and was associated with a decrease in oxidative stress markers, as indicated by elevated levels of ROS and MDA and reduced levels of GSH and SOD. Concurrently, there was an increase in inflammation, as demonstrated by higher levels of IL-6 and TNF-α, and an escalation in apoptosis. This process was accompanied by a dependency on the reduced expression of SNHG17. These findings indicate that our study is the first to confirm SNHG17 as a critical pathway through which DHM exerts its protective actions both *in vitro* and *in vivo*. Given the competitive endogenous RNA (ceRNA) mechanism, SNHG17 may regulate downstream targets by sequestering miRNAs (*Li et al., 2020*), prompting us to identify its potential miRNA targets following confirmation of SNHG17's predominant cytoplasmic localization in BV2 cells *via* nucleocytoplasmic separation experiments.

A key discovery of our study is the validation of miR-452-3p as a potential target of SNHG17, which is downregulated under conditions of OGD/R or MCAO, and upregulated upon intervention with DHM. Although miR-452-3p has been established as a tumorigenic factor in diseases like hepatocellular carcinoma (*Tang et al., 2017*), its

upregulation in microglia correlating with downregulation of CXCR4 expression suggests a potential anti-inflammatory role, yet its function in IS remains elusive. By downregulating SNHG17 expression, DHM potentially impacted miR-452-3p activity since SNHG17 could act as a "molecular sponge" absorbing miR-452-3p, thereby reducing miR-452-3p's suppressive effect on its target CXCR4. Consequently, with decreased SNHG17 expression, miR-452-3p is liberated to more effectively bind and inhibit CXCR4 mRNA, leading to a reduction in CXCR4 protein expression. Multiple studies have confirmed the pro-inflammatory and pro-oxidative roles of CXCR4 in IS or microglial activation (*Yang et al., 2021*; *Kim et al., 2019*). Thus, the SNHG17/miR-452-3p/CXCR4 axis might be pivotal in DHM's antagonism of OGD/R-induced inflammation and oxidative stress, offering fresh insights into how DHM regulates microglial activity and unveiling a potential new therapeutic target for IS treatment. In this study, we observed that DHM's ability to downregulate CXCR4 expression *via* miR-452-3p was accompanied by a concomitant reduction in NF-κB phosphorylation levels. This suggests that CXCR4 serves as an upstream activator of NF-κB, consistent with previous reports highlighting the CXCR4/CXCL12 axis as a significant driver of NF-κB-mediated inflammation in neuroinflammatory conditions.

In the context of IS models, NF-κB, a key transcription factor in inflammatory responses, its activation leads to inflammation and oxidative stress, thereby exacerbating inflammatory injury (*Bhuiyan et al., 2022*; *Xian et al., 2021*). The activity of NF-κB is modulated by a variety of signaling pathways, including upstream receptors, kinases, and inhibitors. Diminished CXCR4 expression can attenuate NF-κB activation as CXCR4's binding with its ligand CXCL12 activates downstream signaling, including NF-κB (*Chai et al., 2022*). Consistent with previous studies, our research observed that DHM indirectly weakens the activation of NF-κB signaling. Specifically, DHM intervention lowered SNHG17 levels, indirectly enhancing miR-452-3p's inhibition of CXCR4, potentially weakening the activation of NF-κB signaling and reducing the post-ischemic inflammatory response, oxidative stress, and apoptosis induced by microglia. Moreover, the activation of the NF-κB signaling is also directly linked to oxidative stress (*Li et al., 2023*). Post-OGD/R, the overproduction of free radicals leads to a cellular stress response, with NF-κB, as a rapid response transcription factor, becoming activated and translocating to the nucleus, inducing the production of inflammatory mediators (*Dong et al., 2021*). Known for its antioxidant properties, DHM might indirectly affect NF-κB activity by ameliorating oxidative stress, further suppressing the inflammatory response (*Awad et al., 2022*). Given that the activation of NF-κB is also a crucial pathway for another form of cell death—pyroptosis—we hypothesized that the inhibition of NF-κB signaling by DHM may also alleviate pyroptosis (*Liu et al., 2023*). The central pathway of pyroptosis involves the NLRP3 inflammasome (with key components NLRP3, ASC, and Caspase 1) facilitating the formation of GSDMD pores on the cell membrane (*Gu et al., 2022*). Previous studies have confirmed that GSDMD pore formation is one of the critical pathways for the release of inflammatory cytokines (*Sun et al., 2023*; *Xu & Nunez, 2023*). This process enables the release of NLRP3-activated pro-inflammatory cytokines IL-1β and IL-18 into the extracellular space, contributing to the formation and exacerbation of the inflammatory

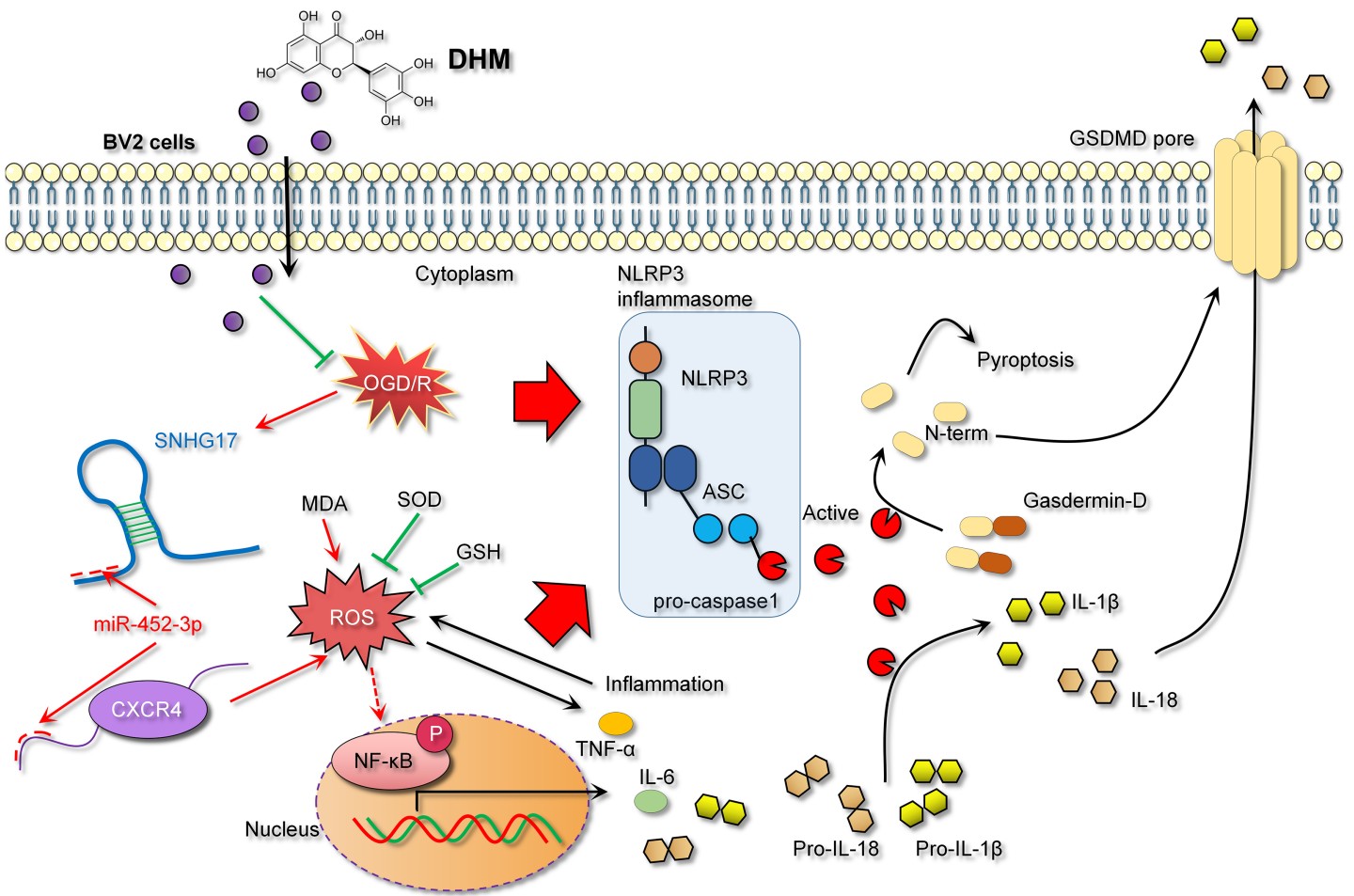

**Figure 10 DHM alleviates the inflammatory environment of IS through the SNHG17/miR-452-3p/CXCR4 axis.**

milieu. In this study, we report that DHM mitigates the activation of the NLRP3 inflammasome induced by OGD/R, leading to the upregulation of NLRP3, ASC, and Caspase 1 expression. This process triggers the formation of GSDMD pores, further promoting the inflammatory environment (Fig. 10).

Despite these promising findings, the limitations of the study must be acknowledged. First, this study employed BV2 cells as an *in vitro* model for ischemic stroke, which cannot fully replicate the complexity of *in vivo* conditions. While the *in vitro* model provides valuable mechanistic insights, it does not account for the dynamic interplay between systemic and local factors in ischemic stroke. Future research should involve detailed animal studies to evaluate the efficacy and safety of DHM and further explore the mechanistic role of the SNHG17/miR-452-3p/CXCR4 molecular axis. While such studies could validate these findings, it is also essential to consider how DHM's effects may vary across different ischemic stroke models or patient subgroups, particularly in relation to genetic diversity or differential miRNA expression profiles. Second, this study did not investigate the effects of DHM or the SNHG17/miR-452-3p/CXCR4 molecular axis in

clinical settings. Such studies could help bridge the gap between cellular findings and clinical applicability. Third, post-ischemic neuroprotection involves a multistep, multifactorial process that extends beyond microglial involvement to include interactions with other brain cell types such as neurons, astrocytes, and endothelial cells. Finally, due to the complexity of ischemic stroke (IS), the therapeutic effects of DHM may involve additional signaling pathways and molecular mechanisms that were not thoroughly investigated in this study. Therefore, a more comprehensive investigation of these interactions, as well as validation in animal models and eventual clinical trials, is necessary to translate our findings into therapeutic practice.

In summary, this study not only provides empirical evidence for the potential of DHM in the treatment of IS but also offers a new perspective on the roles of inflammation and oxidative stress in the pathology of IS. Future research is anticipated to further clarify the impact of DHM on stroke therapy, particularly at the clinical application level. Simultaneously, the role of the SNHG17/miR-452-3p/CXCR4 axis in IS warrants further exploration and study, with the hope of uncovering additional therapeutic targets to bring more treatment options to stroke patients.

### Funding
The work presented in this paper was supported by the Natural Science Foundation of Jiangxi Province (Grant No. 20202BABL206058), the Project of Key Laboratory of Prevention and Treatment of Cardiovascular and Cerebrovascular Diseases, Ministry of Education (Grant No. XN201813), the project of Education Department of Science and Technology of Jiangxi Province (Grant No. 190798), the scientific research project of Gannan Medical University (Grant No. TD201804), the project of Science and Technology Program of Jiangxi Provincial Health Commission (Grant No. 20195407), and the project of Science and Technology of the First Affiliated Hospital of Gannan Medical University (Grant No. YJZD202008). The funders had no role in study design, data collection and analysis, decision to publish, or preparation of the manuscript.

### Grant Disclosures
The following grant information was disclosed by the authors:
Natural Science Foundation of Jiangxi Province: 20202BABL206058.
Project of Key Laboratory of Prevention and Treatment of Cardiovascular and Cerebrovascular Diseases, Ministry of Education: XN201813.
Education Department of Science and Technology of Jiangxi Province: 190798.
Scientific Research Project of Gannan Medical University: TD201804.
Science and Technology Program of Jiangxi Provincial Health Commission: 20195407.
Science and Technology of the First Affiliated Hospital of Gannan Medical University: YJZD202008.

## Competing Interests

The authors declare that they have no competing interests.

## Author Contributions

- Jiacheng Xie conceived and designed the experiments, analyzed the data, authored or reviewed drafts of the article, and approved the final draft.
- Qiuyue Yang performed the experiments, analyzed the data, prepared figures and/or tables, and approved the final draft.
- Xueliang Zeng performed the experiments, analyzed the data, prepared figures and/or tables, and approved the final draft.
- Qi Zeng performed the experiments, analyzed the data, prepared figures and/or tables, and approved the final draft.
- Hai Xiao conceived and designed the experiments, authored or reviewed drafts of the article, and approved the final draft.

## Data Availability

The raw data are available at figshare: Xiao, Hai (2024). Dihydromyricetin inhibits injury caused by Ischemic stroke through the lncRNA SNHG17/miR-452-3p/CXCR4 axis. figshare. Dataset. https://doi.org/10.6084/m9.figshare.26867074.v3.

## Supplemental Information

Supplemental information for this article can be found online at http://dx.doi.org/10.7717/peerj.18876#supplemental-information.

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
