# Peer review of "Dihydromyricetin inhibits injury caused by ischemic stroke through the lncRNA SNHG17/miR-452-3p/CXCR4 axis"

_PeerJ, doi:10.7717/peerj.18876_

## Round 0.1 · original submission · Major Revisions

The primary limitation is the experimental design. The study relies solely on BV2 cell line experiments without in vivo validation or primary microglia confirmation. The selection of DHM concentrations (1-10 μM) lacks pharmacokinetic justification. The OGD/R model parameters require more detailed specification. Additional controls, including scrambled siRNAs and non-targeting miRNA inhibitors, should be included to strengthen the findings' specificity.
The mechanistic investigation needs further development. The relationship between DHM and NF-κB pathway activation requires clarification - whether it's direct or mediated through SNHG17/miR-452-3p/CXCR4. We recommend bioinformatic analysis of other potential miR-452-3p targets and exploration of SNHG17's upstream regulators. Western blot validation of NF-κB nuclear translocation and pyroptosis-related proteins would strengthen the mechanistic claims.
Data presentation requires improvement. Statistical analyses should include exact p-values and sample sizes. Figures 2 and 6 need clearer significance annotations. Figures 6 and 7 lack sufficient quantitative details.
For translational relevance, we strongly recommend including in vivo validation of the SNHG17/miR-452-3p/CXCR4 axis. Additionally, comparison with other neuroprotective agents and discussion of potential variations in DHM effects across different stroke models would enhance clinical significance.

Reviewer 1 ·

Basic reporting

**Basic Reporting**

1. Language and Clarity: The manuscript is written in English; however, the clarity and conciseness of the text can be improved. Some sections would benefit from refining language to enhance readability and professionalism. Certain interpretations, such as those in the results and discussion sections, occasionally use vague terms that could be made more precise by specifying quantitative outcomes or avoiding overly broad descriptors (e.g., “significant suppression” could be clarified with specific data points). Attention to clear and unambiguous phrasing throughout would further align the manuscript with professional standards of scientific reporting.

2. Literature and Background: The introduction provides relevant background on DHM and its potential neuroprotective effects in ischemic stroke; however, it does not fully cover the scope of previous research on DHM’s mechanism, particularly in relation to the SNHG17/miR-452-3p/CXCR4 axis. Additional literature references would strengthen the context, particularly regarding DHM’s known or hypothesized roles in neuroinflammation and oxidative stress, helping readers understand the novelty of the study's focus on this signaling pathway.

3. Figures and Tables: The figures included are relevant to the article’s content but are not all of sufficient resolution. For instance, Figures 2, 5, and 7 have relatively low resolution, which impacts the clarity of the data presented, particularly in terms of axes, labels, and specific data points. Additionally, certain figure legends lack comprehensive descriptions, making it difficult for readers to independently interpret the data without cross-referencing the text. Improving the quality and detail of figures and legends would significantly enhance the paper’s presentation quality.

4. Self-Containment and Hypothesis-Relevant Results: The manuscript is largely self-contained, presenting data that aligns with its hypothesis. However, certain key elements, such as the methodological justification for chosen DHM concentrations and OGD/R times, are not fully explained, which can limit the manuscript's completeness. Providing a rationale for experimental settings and ensuring that the results directly support the proposed hypotheses would strengthen the study’s cohesiveness and self-containment.

Overall, the manuscript is appropriately structured but could benefit from enhancements in language precision, data presentation, and literature depth to ensure clarity, alignment with professional standards, and adherence to data-sharing requirements.

Experimental design

Relevance to Journal Scope and Research Question: The study presents original research within the field of neuroprotection and ischemic stroke, aligning well with the journal's focus. The research question—investigating DHM’s role in neuroprotection via the SNHG17/miR-452-3p/CXCR4 axis—is clear, relevant, and addresses a meaningful gap in understanding the molecular mechanisms of DHM in ischemic stroke. However, the study could further highlight how its findings advance this field specifically and where it contributes beyond existing research on DHM’s anti-inflammatory and antioxidant properties.

Rigor and Technical Standards: The experimental setup is generally rigorous, with various assays (MTT, qPCR, ELISA) used to evaluate cell viability, gene expression, and cytokine levels. However, a few aspects could be improved for greater rigor:
Control Groups: Adding more comprehensive control groups, especially a non-DHM-treated OGD/R group, would strengthen conclusions regarding DHM’s specific effects.
Multiple Comparisons and Replicates: The study does not clarify whether statistical corrections for multiple comparisons (e.g., Bonferroni correction) were applied, nor does it specify the number of biological and technical replicates for key experiments. Clarifying these elements would improve statistical reliability and align with high standards in experimental rigor.
Ethical Standards: The study uses BV2 microglial cells as an in vitro model for ischemic stroke. While ethical considerations are minimal for in vitro studies, the manuscript could discuss the model’s limitations in relation to in vivo complexity and suggest future directions for validating findings in animal models. This would enhance transparency regarding the study’s translational limitations.

Methodological Detail and Replicability: Overall, methods are sufficiently detailed, but a few areas require additional information for full reproducibility:
DHM Concentrations: The rationale for selecting 1, 5, and 10 µM concentrations of DHM is not provided. Including a justification, ideally supported by prior literature or pilot data, would clarify these methodological choices.
qPCR Primer Sequences: While qPCR primers are referenced, specific sequences for SNHG17, miR-452-3p, and other targets are not included. Providing these sequences and any validation data would be essential for replicability.
OGD/R Model Parameters: Details on the oxygen-glucose deprivation and reoxygenation (OGD/R) protocol are minimal, especially regarding the duration and oxygen levels used. Expanding this description with references to existing models would allow other researchers to accurately reproduce the setup.

Validity of the findings

Data Robustness and Statistical Soundness: The study’s data are compelling and generally align with the stated research question regarding DHM’s neuroprotective effects via the SNHG17/miR-452-3p/CXCR4 pathway. However, there are areas where data robustness could be improved:
Multiple Comparisons: Although p-values are reported, it is unclear whether any corrections were applied for multiple comparisons across experiments. Implementing statistical corrections, such as Bonferroni adjustments, would enhance the soundness of the findings and support confidence in the reported significance levels.


Replication of Results: Information on biological and technical replicates is not clearly specified in the methods section. Ensuring that all experiments include adequate replication, ideally with triplicates, is essential for robust conclusions.
Control and Consistency of Findings:
The study presents a range of data supporting DHM’s effects on cell viability, inflammation, and oxidative stress markers. However, the inclusion of additional control groups (e.g., untreated OGD/R) would provide a clearer baseline, making it easier to attribute observed effects specifically to DHM rather than to experimental variability.
Figures, particularly those measuring cytokines and oxidative stress markers (e.g., ROS, MDA), show some variability. Explaining potential causes of data variability and indicating any outlier handling approaches would add to the study’s transparency and the reliability of its conclusions.

Linkage Between Conclusions and Data: The conclusions are generally well-stated, focusing on DHM’s regulatory effect on the SNHG17/miR-452-3p/CXCR4 axis. However, a few aspects could benefit from refinement to ensure conclusions are closely tied to the data:
Specificity of Pathway Conclusions: The study’s conclusions regarding the SNHG17/miR-452-3p/CXCR4 axis could be better supported by additional pathway-specific controls, such as rescue experiments or specific pathway inhibitors. Including such controls would solidify the claim that DHM’s neuroprotective effects are indeed mediated through this specific pathway.
Quantitative Detail in Results Interpretation: In some cases, interpretations are qualitative rather than quantitative (e.g., "significant suppression"). More quantitative statements, such as percent reductions or fold-changes, would provide stronger links between the findings and the conclusions.
Data Sharing and Transparency: The manuscript does not specify whether all underlying raw data are available, particularly for critical measurements like qPCR and ELISA data. To meet data-sharing standards, it is recommended that the authors clarify the availability of these data to facilitate meaningful replication by other researchers and enhance the transparency of the study.

Additional comments

Introduction - Reference to Previous Work (Lines 44–92)
Provide references to previous studies showing DHM’s effects on inflammatory pathways in neural contexts to better support its potential role in ischemic stroke, especially relating to SNHG17/miR-452-3p/CXCR4 signaling.

Details on DHM Treatment (Lines 95–104)
Clarify why concentrations of 1, 5, and 10 µM for DHM were selected, and whether previous dose-response findings in similar cell models informed these choices. This additional context could enhance the rationale for the dosing strategy.

Biological and Technical Replicates (Lines 110–119)
Specify the number of biological and technical replicates performed in each key assay, such as MTT and ELISA. For example, indicate if three independent experiments with triplicates per condition were used to ensure the robustness of findings.

Resolution and Labeling of Figures 2, 5, and 7 (Figures)
The resolution of Figures 2B, 5A, and 7C is insufficient for detailed examination. Additionally, consider improving figure legends to clarify experimental conditions and controls, particularly in the bar graph representations.

Discussion on NF-κB Pathway Role (Lines 319–327)
Further expand on NF-κB’s involvement in the DHM-mediated pathway. Adding relevant citations and a more detailed discussion of NF-κB’s influence on the SNHG17/miR-452-3p/CXCR4 axis would enrich this section.

Primer Sequence Details for Reproducibility (Table 1)
Include primer sequences for SNHG17, miR-452-3p, and CXCR4 to improve transparency and enable reproducibility. Also, report primer efficiencies and validation metrics if available.

OGD/R Model Justification (Lines 98–101)
Explain the choice of OGD duration and reoxygenation times, citing previous studies on ischemic stroke cell models. This will strengthen the experimental validity and rationale for using the OGD/R setup in BV2 cells.

Specificity of SNHG17/miR-452-3p Binding (Lines 220–232)
Confirming the specificity of the SNHG17 and miR-452-3p interaction via site-directed mutagenesis or additional controls (e.g., using mutated target sequences) could strengthen this mechanistic aspect of the study.

Limitations of Using Only Microglial BV2 Cells (Lines 338–340)
Discuss the translational limitations of using BV2 microglial cells alone. Suggest that future studies include co-cultures with neurons and astrocytes or validation in animal models to better emulate the in vivo microenvironment.

Inclusion of Additional Control Group (Lines 265–267)
For cytokine assays, a non-DHM-treated OGD/R control would help distinguish DHM-specific effects on inflammatory responses and strengthen the interpretation of results in inflammatory cytokine modulation.

Reviewer 2 ·

Basic reporting

The manuscript "Dihydromyricetin inhibits injury caused by Ischemic stroke through the lncRNA SNHG17/miR-452-3p/CXCR4 axis" presents intriguing data on the protective effects of Dihydromyricetin (DHM) in ischemic stroke models, highlighting a complex molecular mechanism involving the SNHG17/miR-452-3p/CXCR4 regulatory axis. This research aligns well with current interest in neuroprotective agents targeting inflammation and oxidative stress in ischemic stroke. However, several areas warrant clarification and enhancement to bolster the manuscript's impact and scientific rigor.
1.The manuscript should further emphasize DHM’s uniqueness compared to other neuroprotective agents in ischemic stroke models. Consider discussing existing studies on alternative flavonoids or antioxidants and contrast their mechanisms with DHM’s to underscore its distinctiveness in regulating the SNHG17/miR-452-3p/CXCR4 axis.
2.While the authors have demonstrated that DHM acts through SNHG17, miR-452-3p, and CXCR4, more depth in the mechanistic pathways is needed, particularly regarding CXCR4's downstream effects on NF-κB. Including additional Western blot or immunofluorescence data on NF-κB nuclear translocation and target gene expression would reinforce the proposed signaling pathway. Additionally, examining upstream regulators of SNHG17 in the OGD/R model could add another dimension to the pathway analysis.
3.Clarification is needed on the rationale for selecting specific DHM concentrations. It would benefit readers to know if these concentrations have been validated in previous in vitro or in vivo ischemic models. This would lend credence to the chosen range and help readers understand the clinical relevance of the DHM doses used.
4.The study's in vitro model of ischemia, while informative, lacks in vivo validation. Including a section that discusses how well the findings might translate to in vivo or clinical contexts—potentially supported by future studies in animal models—would enhance the relevance of the findings for stroke treatment.
5.The regulatory network involving SNHG17 and miR-452-3p appears complex. As miR-452-3p might target multiple genes beyond CXCR4, providing a bioinformatics analysis of additional targets or cross-verifying with RNA pull-down results could strengthen the conclusions and address any off-target effects that might confound the observed results.
6.The statistical treatment in some figures lacks clarity. For example, Figures 2 and 6 would benefit from an annotation of p-values directly on the graphs to clearly indicate significant differences between conditions. Additionally, including exact sample sizes (n-values) for each experiment would improve reproducibility.
7.While the manuscript acknowledges the in vitro limitations, it would benefit from a more thorough discussion on how DHM’s effect might vary across different ischemic stroke models or patient subgroups. Consider suggesting future research on personalized approaches to using DHM based on genetic profiles or miRNA expression levels.
8.Some areas of the text, particularly in the results section, would benefit from clearer, more concise language. Consider reducing redundancy, particularly in descriptions of experimental results, and focusing on interpreting the findings rather than reiterating raw data points.
9.The current introduction provides extensive general information on ischemic stroke and microglial cells. It is recommended to condense the background on ischemic stroke and the dual role of microglia in inflammation within the initial paragraphs, using concise language. This approach would expedite the transition to the core topic of this study—the potential mechanisms of DHM—avoiding excessive general information that may dilute the reader’s focus.
10.It is advisable to introduce the primary scientific question of the study early in the introduction and to clarify the study’s innovative aspects. For instance, while DHM’s anti-inflammatory and antioxidant properties as a flavonoid have been previously reported, its precise mechanisms in ischemic injury remain unclear. Emphasizing the novel regulatory role of the SNHG17/miR-452-3p/CXCR4 pathway in this context would underscore the study’s uniqueness.
11.Reduce citations of non-core literature, focusing instead on key studies: The current references cover a broad scope; it would be beneficial to prioritize literature that directly supports the mechanisms of DHM, limiting citations that merely provide background. Concentrating on essential studies related to SNHG17, miR-452-3p, CXCR4, and inflammation regulation would make the introduction more streamlined and focused.
12.Although the introduction mentions research on SNHG17 and miR-452-3p, their specific regulatory roles in inflammation and oxidative stress have not been fully explained. Including additional literature that supports SNHG17’s function as a ceRNA regulating miR-452-3p and influencing CXCR4 expression would provide a theoretical foundation for the subsequent experiments.
13.Toward the end of the introduction, a brief mention of DHM’s potential clinical applications as a plant-derived compound for ischemic stroke is recommended. This section could summarize the study’s clinical significance and highlight how this research addresses a knowledge gap in the mechanisms of DHM, further emphasizing the importance of the study.
14.The oxygen-glucose deprivation/reoxygenation (OGD/R) model description lacks specific details regarding the duration and precise conditions for each phase (OGD and reoxygenation). For reproducibility, specify the exact timing of hypoxia and reoxygenation steps, including gas composition in the hypoxic chamber and temperature. Additionally, it would be helpful to mention whether cell viability was assessed immediately following reoxygenation or after a recovery period.
15.Given the complexity of the SNHG17/miR-452-3p/CXCR4 axis, it is essential to verify the specificity of observed interactions. Consider including negative control experiments, such as using scrambled siRNAs or non-targeting miRNA inhibitors, to strengthen the specificity of SNHG17’s and miR-452-3p’s effects. These controls would enhance the credibility of the conclusions and address potential off-target effects.
16.Many sentences contain redundant language that can be streamlined for clarity and conciseness. For instance, phrases like “the results of our study show that…” could be simplified to “Our results show that…”. Removing unnecessary words will make the narrative more direct and scientifically rigorous, allowing the core findings to be more prominent.
17.There are several instances where modifier placement could be adjusted to avoid ambiguity. For example, instead of “DHM significantly enhanced cell viability in OGD/R-induced cells in a dose-dependent manner,” consider “In OGD/R-induced cells, DHM enhanced cell viability in a dose-dependent manner.” Placing modifiers closer to the subjects they describe will prevent misinterpretation and ensure precision in describing experimental outcomes.

Experimental design

See Basic reporting.

Validity of the findings

See Basic reporting.

Reviewer 3 ·

Basic reporting

The manuscript investigates the neuroprotective role of dihydromyricetin (DHM) in ischemic stroke (IS) using an in vitro model. The authors demonstrate that DHM attenuates oxygen-glucose deprivation/reoxygenation (OGD/R)-induced injury in BV2 microglial cells through the regulation of the lncRNA SNHG17/miR-452-3p/CXCR4 axis. Their findings suggest that DHM downregulates SNHG17, liberating miR-452-3p to inhibit CXCR4 expression, thereby reducing inflammation and oxidative stress via the NF-κB signaling pathway. Additionally, DHM mitigates inflammasome activation and pyroptosis. The results provide insights into the potential of DHM as a therapeutic agent for ischemic stroke. While the study addresses a relevant scientific gap and uses robust experimental techniques, certain methodological and interpretative aspects require clarification.

Experimental design

While BV2 microglia are a suitable model for preliminary studies, the relevance of findings to primary microglia or in vivo ischemic stroke models should have been included. Are there differences in the SNHG17/miR-452-3p/CXCR4 axis between in vitro and in vivo systems?

Validity of the findings

The selection of DHM concentrations (1–10 µM) is not adequately justified. Including pharmacokinetic data to substantiate the relevance of these concentrations would improve the physiological validity of the study.
The results in Fig 9 regarding pyroptosis-related markers should be validated using either western blot or ELISA techniques for increased robustness.

Additional comments

The manuscript investigates the neuroprotective role of dihydromyricetin (DHM) in ischemic stroke (IS) using an in vitro model. The authors demonstrate that DHM attenuates oxygen-glucose deprivation/reoxygenation (OGD/R)-induced injury in BV2 microglial cells through the regulation of the lncRNA SNHG17/miR-452-3p/CXCR4 axis. Their findings suggest that DHM downregulates SNHG17, liberating miR-452-3p to inhibit CXCR4 expression, thereby reducing inflammation and oxidative stress via the NF-κB signaling pathway. Additionally, DHM mitigates inflammasome activation and pyroptosis. The results provide insights into the potential of DHM as a therapeutic agent for ischemic stroke. While the study addresses a relevant scientific gap and uses robust experimental techniques, certain methodological and interpretative aspects require clarification.
Comments:
1. While BV2 microglia are a suitable model for preliminary studies, the relevance of findings to primary microglia or in vivo ischemic stroke models should have been included. Are there differences in the SNHG17/miR-452-3p/CXCR4 axis between in vitro and in vivo systems?
2. The role of the NF-κB pathway is discussed, but additional experiments that could clarify whether it is directly modulated by DHM or indirectly through SNHG17/miR-452-3p/CXCR4 are lacking. Have off-target effects of DHM on other signaling pathways been ruled out?
3. Incorporate an in vivo ischemic stroke model to validate the SNHG17/miR-452-3p/CXCR4 axis and confirm the therapeutic potential of DHM.
4. The selection of DHM concentrations (1–10 µM) is not adequately justified. Including pharmacokinetic data to substantiate the relevance of these concentrations would improve the physiological validity of the study.
5. In Fig 4D, the authors excluded some miRNAs using established mechanistic studies in brain diseases via PubMed but did not provide the exclusion criteria. This raises concerns about selection bias.
6. Figures 6 and 7 lack quantitative details and clear captions. All data points in graphs should be included to enhance transparency and reproducibility.
7. The results in Fig 9 regarding pyroptosis-related markers should be validated using either western blot or ELISA techniques for increased robustness.
8. The authors should comment on whether GSDMD pore formation is the only or critical pathway through which inflammatory cytokines are released from cells under OGD/R conditions.
9. The discussion could be strengthened by including comparisons between DHM and other antioxidant or anti-inflammatory agents, and by situating findings within the broader context of existing research on ischemic stroke.
10. Regarding the language quality, some sentences are overly verbose and could be condensed for clarity.

---

## Round 0.2 · accepted · Accept

Since authors have fully revised according to comments, I think this paper can be accepted.

Reviewer 1 ·

Basic reporting

The author's revision is comprehensive and meticulous, and I think the revised manuscript can be accepted.

Experimental design

no comment

Validity of the findings

no comment

Reviewer 2 ·

Basic reporting

The revision is valid

Experimental design

The revision is valid

Validity of the findings

The revision is valid

Reviewer 3 ·

Basic reporting

The revision is satisfactory.

Experimental design

The revision is satisfactory.

Validity of the findings

The revision is satisfactory.

Additional comments

The revision is satisfactory.